# Asian dust-storm activity dominated by Chinese dynasty changes since 2000 BP

Fahu Chen[1,2,3,9 ✉], Shengqian Chen[1,9], Xu Zhang[1,3,4], Jianhui Chen [1], Xin Wang [1], Evan J. Gowan [4], Mingrui Qiang[1,5], Guanghui Dong[1], Zongli Wang[1], Yuecong Li[6], Qinghai Xu[6], Yangyang Xu[7], John P. Smol [8] & Jianbao Liu[1,2,3 ✉]

The Asian monsoon (AM) played an important role in the dynastic history of China, yet it remains unknown whether AM-mediated shifts in Chinese societies affect earth surface processes to the point of exceeding natural variability. Here, we present a dust storm intensity record dating back to the first unified dynasty of China (the Qin Dynasty, 221–207 B.C.E.). Marked increases in dust storm activity coincided with unified dynasties with large populations during strong AM periods. By contrast, reduced dust storm activity corresponded to decreased population sizes and periods of civil unrest, which was co-eval with a weakened AM. The strengthened AM may have facilitated the development of Chinese civilizations, destabilizing the topsoil and thereby increasing the dust storm frequency. Beginning at least 2000 years ago, human activities might have started to overtake natural climatic variability as the dominant controls of dust storm activity in eastern China.

[1] Key Laboratory of Western China's Environmental Systems (Ministry of Education), College of Earth and Environmental Science, Center for Pan Third Pole Environment (Pan-TPE), Lanzhou University, Lanzhou 730000, China. [2] Key Laboratory of Alpine Ecology (LAE), Institute of Tibetan Plateau Research, Chinese Academy of Sciences (CAS), Beijing 100101, China. [3] CAS Center for Excellence in Tibetan Plateau Earth Sciences, Chinese Academy of Sciences (CAS), Beijing 100101, China. [4] Alfred Wegener Institute Helmholtz Centre for Polar and Marine Research, D-27570 Bremerhaven, Germany. [5] School of Geography Sciences, Guangdong Provincial Center for Smart Land Research, South China Normal University, Guangzhou 510631, China. [6] Institute of Nihewan Archaeology Research, College of Resources and Environment, Hebei Normal University, Shijiazhuang 050024, China. [7] Department of Atmospheric Science, Texas A&M University, College Station 77842 TX, USA. [8] Paleoecological Environmental Assessment and Research Lab (PEARL), Department of Biology, Queen's University, Kingston, ON K7L 3N6, Canada. [9] These authors contributed equally: Fahu Chen, Shengqian Chen. ✉email: fhchen@lzu.edu.cn; jbliu@itpcas.ac.cn

Changes in monsoon rainfall have played a key role in the cultural development of many regions, such as the collapse of the Maya[1], the recession of the Indus Civilization[2], and the development of different dynasties in China[3,4]. Asian monsoon (AM) rainfall accounts for up to 80% of total precipitation in China. Transitions between Chinese dynasties tended to occur during periods of a weak AM, as the associated reduction in monsoon rainfall resulted in crop failure and famine, leading to civil unrest and warfare[3–5]. In contrast, agricultural expansion and dramatic increases in crop production typically coincided with intensified rainfall during periods of a strong AM[3,5]. This was the basis for unified dynasties that were characterized by sharp increases in population (in the tens of millions) within only a few decades[6], such as the Western Han, Tang and Northern Song dynasties[3–5].

The history of the past AM, as well as the dynastic history of China, is now reasonably well established[3,7], but the ecological effects of AM-mediated cultural changes remain largely unknown. Moreover, it remains unclear whether the rapid increases in population could generate landscape changes that exceeded natural variability[8,9]. To address these questions, a reliable proxy record regarding landscape changes is needed to compare with the history of the AM and the dynastic history of China.

Northern China has long been recognized as the center of Chinese civilization, and it was one of the most densely populated areas of ancient China[6]. Since its location is close to the boundary of the AM system (Fig. 1d), the ecological environment is very sensitive to movements of the monsoon rainfall belt. The AM-mediated shifts in Chinese society were consistently accompanied by large population fluctuations in this area[5,6]. Like other civilization centers in the world, population increases may lead to extensive landscape modification due to the expansion of agriculture[10–12] and desertification[13]. These disturbed earth surface processes can potentially exert broader impacts on socioeconomic and ecological systems downwind in eastern China.

Information archived in dated lake sediments can be used to reconstruct past climatic changes[1,4] and to infer the environmental impacts of ancient civilizations[14]. Lake Gonghai (38°54′ N, 112°14′ E; 1840 m above mean sea level) was strategically chosen for this study because it is a hydrologically-closed alpine lake located on the Chinese Loess Plateau (Supplementary Fig. 1a, c). The Chinese Loess Plateau contains the largest and thickest dust deposits on Earth, covering a total area of over 640,000 km$^2$ (ref. [15]). Lake Gonghai has a small, undisturbed catchment with a surface area of 0.36 km$^2$, a maximum water depth of 10 m, and a flat bottom topography. Being a remote and high-altitude lake, Lake Gonghai is undisturbed by local human activity, and therefore is ideal for

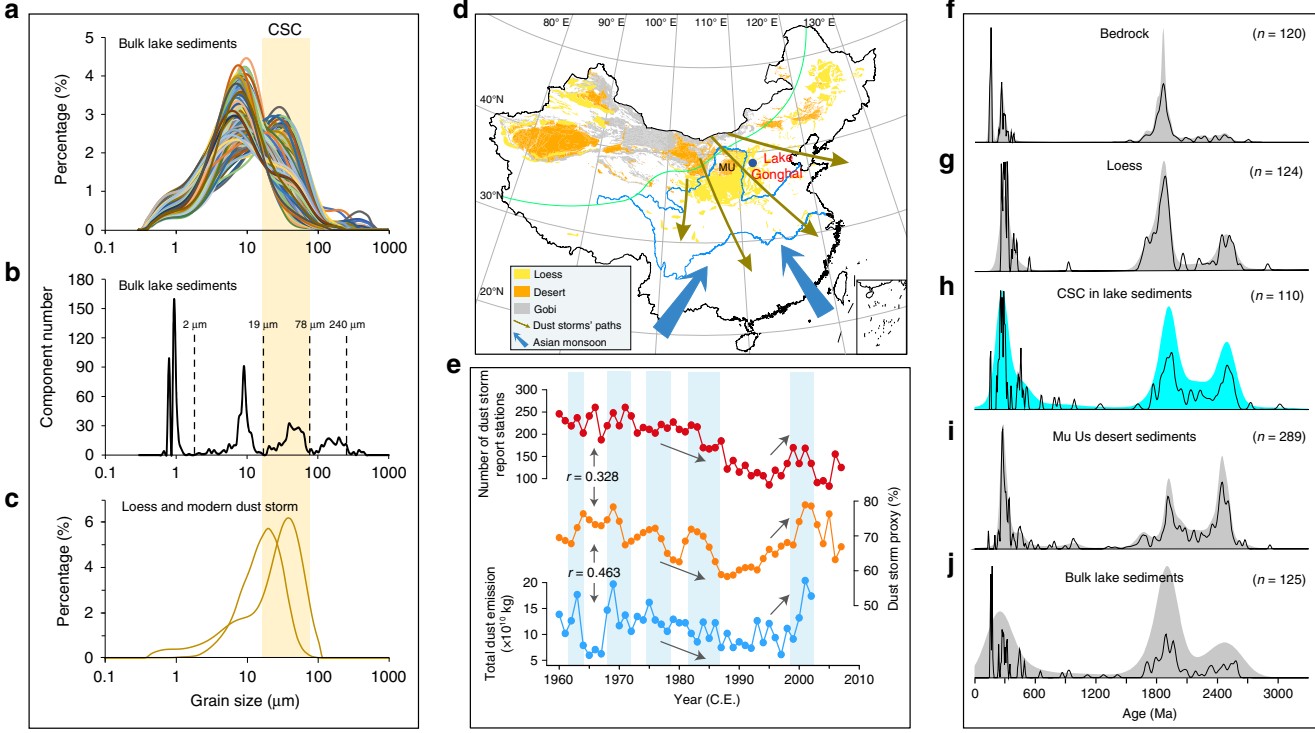

**Fig. 1 Establishment of a dust storm proxy in the sediments of Lake Gonghai. a** Measured grain-size distributions of 340 samples from core GH09B. **b** Frequency distribution of the component size of the sediments of core GH09B using the grain-size distribution function method[19]. From this analysis, a distinguishable coarse silt component is identified, indicated by the yellow shading. **c** Measured grain-size distributions of surrounding loess and modern dust storms deposits[44]. **d** Distributions of loess (yellow shading)[15], desert (orange shading), and Gobi (gray shading) (http://westdc.westgis.ac.cn/) in China. The modern AM limit is shown by the solid green line. Locations of Lake Gonghai (blue dot), the Mu Us Desert (MU), and the trajectories of dust storms (yellow–brown arrows) and the AM (blue arrows) are indicated. **e** Reconstructed dust storm variations (yellow curve) based on the CSC$_{13F}$ and its relationship with monitored dust storm frequency (red curve) and estimated dust emission (blue curve)[26]. Station numbers of dust storms in China are based on ref. [26] and the China National Meteorological Data Service Center (http://data.cma.cn/). The resolution of the grain-size data were transformed to a 1 year/interval using linear interpolation, for comparison with the meteorological and simulated data. The blue shading corresponds to periods of substantial change in reconstructed dust storm activity, and the gray arrows indicate the main trends. The Pearson's correlation coefficient between the dust storm proxy and meteorological and simulated data is also shown. **f–j** Zircon U–Pb dating results of bedrock (**f**), surrounding loess (**g**), the isolated CSC in lake sediments (**h**), surface sediments of the Mu Us Desert[20] (**i**), and bulk lake sediments (**j**). The black lines and shaded area represent probability density and kernel density estimation plots, respectively. Panel **d** was created using Arcmap 10.2.

studying large-scale environmental change caused by both climate change and human activity[7,16]. Grain-size analysis of lake sediments is widely used to reconstruct large-scale environmental changes caused by human activity[17,18]. The dynamics of different sediment transport processes can be quantitatively separated by numerical partitioning of grain-size components[19].

Here, we present four independent lines of evidence which confirm that the coarse silt component (CSC) from the sediments of Lake Gonghai is an indicator of past dust storms. The variation of the percentages of the CSC from a [14]C-dated, long high-resolution sediment core (GH09B, ~5 years/interval) provides robust evidence for synchronous changes in dust storm activity, recorded population, and monsoon rainfall. Sharp increases in dust storm activity corresponded to unified dynasties with large populations during periods of strengthened AM, while reduced dust storm frequency coincided with a decreased population, civil unrest, and a weakened AM. This suggests human activities, beginning at least 2000 years ago, began to overtake natural climatic variability as the dominant control of dust storm activity in eastern China.

## Results

**Determination of a dust storm indicator.** Based on our observations (Supplementary Fig. 2a), there are two main sources of sediments in Lake Gonghai: one is derived from surface runoff from the catchment, and the other is eolian deposition derived from the surrounding arid and semiarid regions. Sediments of different provenances in Lake Gonghai have totally different grain-size characteristics. The grain-size distributions of runoff-transported materials that are sourced from weathered catchment bedrock are dominated by sand (Supplementary Fig. 2f), whereas the regional eolian sediments are dominated by coarse silt (Supplementary Fig. 2c, d). The nonoverlapping grain-size distributions can distinguish the provenances of different materials in the lake sediments.

Using the grain-size distribution function method[19], five grain-size components were separated from two types of sediment: the surface lake sediments and the sediments from the long (GH09B) and short (GH13F) cores. Based on their modal size, the five components are defined as clay, fine silt, coarse silt, fine sand, and sand (Supplementary Figs. 3–5). As discussed later, the coarse silt component (CSC, modal size range of 19–78 μm) is a robust proxy for dust storm activity, supported by four independent lines of evidence in the following.

First, except for the CSC in the sediments of core GH09B, all of the components are transported by local surface runoff to Lake Gonghai (see Methods). This allows the CSC to be used as a proxy for reconstructing eolian processes, which reflect large-scale changes in the regional environment. The percentage of $CSC_{surface}$ (the CSC in surface lake sediments) is characterized by an irregular distribution along a profile spanning the long axis of the lake (Supplementary Fig. 3d, blue line), demonstrating that the CSC is not transported to the lake center by surface runoff from bedrock debris and eolian deposition within the lake catchment.

Second, U–Pb dating of zircon grains isolated from the CSC in sediment core GH09B reveals three dominant peaks, with ages of ~300, 1900, and 2500 Ma (Fig. 1h), which are well matched with those present in the nearby loess (which is a product of past dust storms) (Fig. 1g). Moreover, the U–Pb dating results are consistent with those from surface sediments of the Mu Us Desert and the surrounding sandy lands[20,21] (Fig. 1i), which are the modern center of dust storm activity in China[22]. However, the U–Pb ages of bedrock from the lake catchment and bulk lake sediments are dominated by two main peaks, with ages of 150 and 1900 Ma, and

they lack a peak at 2500 Ma (Fig. 1f, j). This pattern is different from that of the CSC and the surface sediments of the Mu Us Desert. Therefore, the CSC mainly originates from the proximal desert and sandy lands in northern China, rather than from within the lake catchment.

Third, the grain-size distributions of the surrounding loess and modern dust storms (Fig. 1c) are consistent with those of the CSC (Fig. 1a, b). Moreover, the grain-size distributions of the CSC in the sediments of core GH09B are consistent with those of isolated quartz grains from the same sediments (Supplementary Fig. 6a), which are characterized by eolian surface features (Supplementary Fig. 6b–f). These features include dish-shaped concavities, mechanically-formed upturned plates, elongated depressions, smooth precipitation surfaces, cleavage faces, and arcuate fractures (Supplementary Fig. 6b–f). Although these features may be subsequently affected by other processes, the original grain-size properties would be largely retained, indicating eolian transport[23–25].

Fourth, based on the percentage of $CSC_{13F}$ (the CSC in sediment of core GH13F) of the [210]Pb-dated short core, the variations in dust storm activity over the past five decades exhibit interannual oscillations before 1985 C.E., a rapid decrease in the 1990s, and an abrupt increase at ~2000 C.E (Fig. 1e, yellow line). The significant correlation of reconstructed dust storm activity with observed dust storm frequency[26] ($r = 0.328$, $p < 0.05$), and with simulated Asian dust emissions[26] ($r = 0.463$, $p < 0.005$), further supports an eolian origin of the CSC (Fig. 1e).

Overall, these four independent lines of evidence corroborate our conclusion that the percentage of the CSC in the Lake Gonghai sediments is an unambiguous indicator of dust storm activity over eastern China. This allows us to reconstruct past dust storm variability which is regarded as a major factor influencing the wellbeing of the hundreds of millions of people downwind in eastern China[15,22].

**Multidecadal Asian dust storm variation during the past 2250 years.** Our paleolimnological record extends the history of dust storm activity back to 250 B.C.E. Multidecadal dust storm activity is characterized by multiple abrupt shifts during the past 2250 years (Fig. 2c). Pronounced increases in dust storm activity occurred at the beginning and middle of unified dynasties (Fig. 2c), such as the Western Han, Tang, Northern Song, and Ming, when agricultural productivity was expanded[5,11]. Enhanced dust storm activity (Fig. 2c) can be closely linked to population fluctuations in the dust source region[6] (Fig. 2b), and has a strong temporal coherence with increased AM rainfall[7] (Fig. 2d). In contrast, periods of decreased dust storm activity are coeval with periods of declining population in the dust source region, corresponding to times of decreased agricultural productivity. In particular, dust storm activity decreased substantially during periods of civil unrest, such as the Era of Disunity (220–589 C.E.), the Five Dynasties and Ten Kingdoms period (5D and 10K, 907–979 C.E.), and the final stages of the Eastern Han, Tang, Northern Song, and Ming dynasties (Fig. 2).

**Asian dust-storm activity dominated by Chinese dynastic changes.** Dust storms are caused by wind erosion of surface soil. Variations in dust storm activity are therefore associated with both climatic factors (surface wind speed, precipitation)[15,22,27] and human activity[28]. A humid climate, with higher monsoon rainfall, tends to maintain soil moisture levels, promoting vegetation growth and weakening dust storms[15,27]. In contrast, dust storm activity can be intensified by strong winds and grassland degradation caused by reduced rainfall and expanded cultivation in the dust source region[17,22,28]. Our records show that pronounced

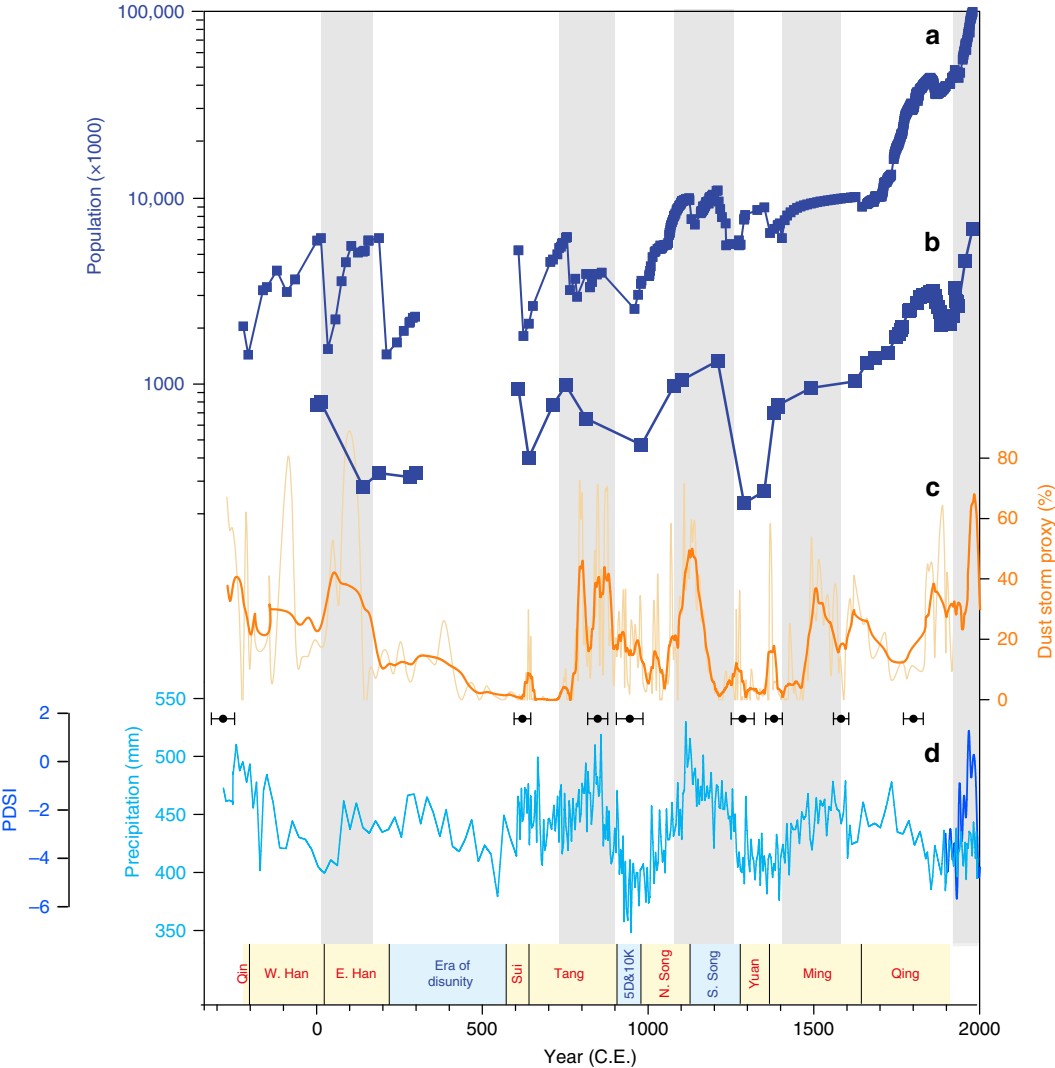

**Fig. 2 Comparison of the dust storm record with variations in population and monsoon rainfall.** Population variations in China[6] (**a**) and in the dust source region (i.e., Shanxi, Shaanxi, Ningxia, and Inner Mongolia)[6], which are defined by zircon U–Pb dating (**b**). During the Era of Disunity, there was long-term civil unrest, during which no national demographic data were recorded in the contemporary literature[6]. **c** Reconstructed dust storm variations from Lake Gonghai (light-yellow raw data smoothed with a dark-yellow five-point moving average). The raw radiocarbon age-control points with error bar are indicated (Supplementary Table 1). **d** AM rainfall record[7] after removing the long-term trend of more than 500 years (sky-blue line), and an AM index based on tree-ring data from the Chinese Loess Plateau[47] (dark blue line). Gray shading corresponds to periods of substantially increased dust storm activity, which have a strong temporal coherence with the evolution of population and monsoon rainfall. Chinese dynasties since the Qin are indicated at the bottom.

increases in dust storm activity (Fig. 2c) correspond to more humid periods with enhanced AM rainfall in northern China (Fig. 2d) and are unrelated to variations in wind strength[29] (Supplementary Fig. 7). This suggests that climatic factors played a limited role in modulating dust storm activity. In contrast, abrupt changes in dust storm activity on multidecadal to centennial timescales (Fig. 2c) are coincident with dramatic fluctuations in population (Fig. 2b). The increasing population since the Han Dynasty[6], advances in agricultural technology[11] and the associated deforestation and desertification substantially impacted the landscape and environment. The close correspondence between population size and dust storms suggests that human activity was the dominant factor influencing changes in dust storm activity. In addition, pronounced increases in human activity (Fig. 2a) exhibit a strong temporal coherence with the unified dynasties (bottom bar in Fig. 2). These indicate that dynastic changes dominated dust storm activity in eastern China since 2000 B.P.

## Discussion

A strengthened monsoon shifts the rainfall belt northwards in northern China[30], increasing crop yields and facilitating the development of unified dynasties with high agricultural production[1,5]. Enhanced monsoon rainfall enabled the expansion of cultivation into previously unfavorable areas, which promoted large population increases in the monsoon marginal zone. It has been documented that the farming-pastoral ecotone shifted substantially northwards during intervals of increased monsoon rainfall, which was accompanied by the development of cities during the Han, Tang, Northern Song, and Ming dynasties[13,31,32] (Fig. 3a, c, d, f). The dense populations in turn prompted the expansion of cultivation, grassland degradation, and exposure of the surface soil to wind erosion[10–12]. These processes increased the amount of sediment available for eolian transport[33], resulting in an abrupt increase in dust storm activity. In contrast, the abrupt population decreases in the dust source region, corresponding to

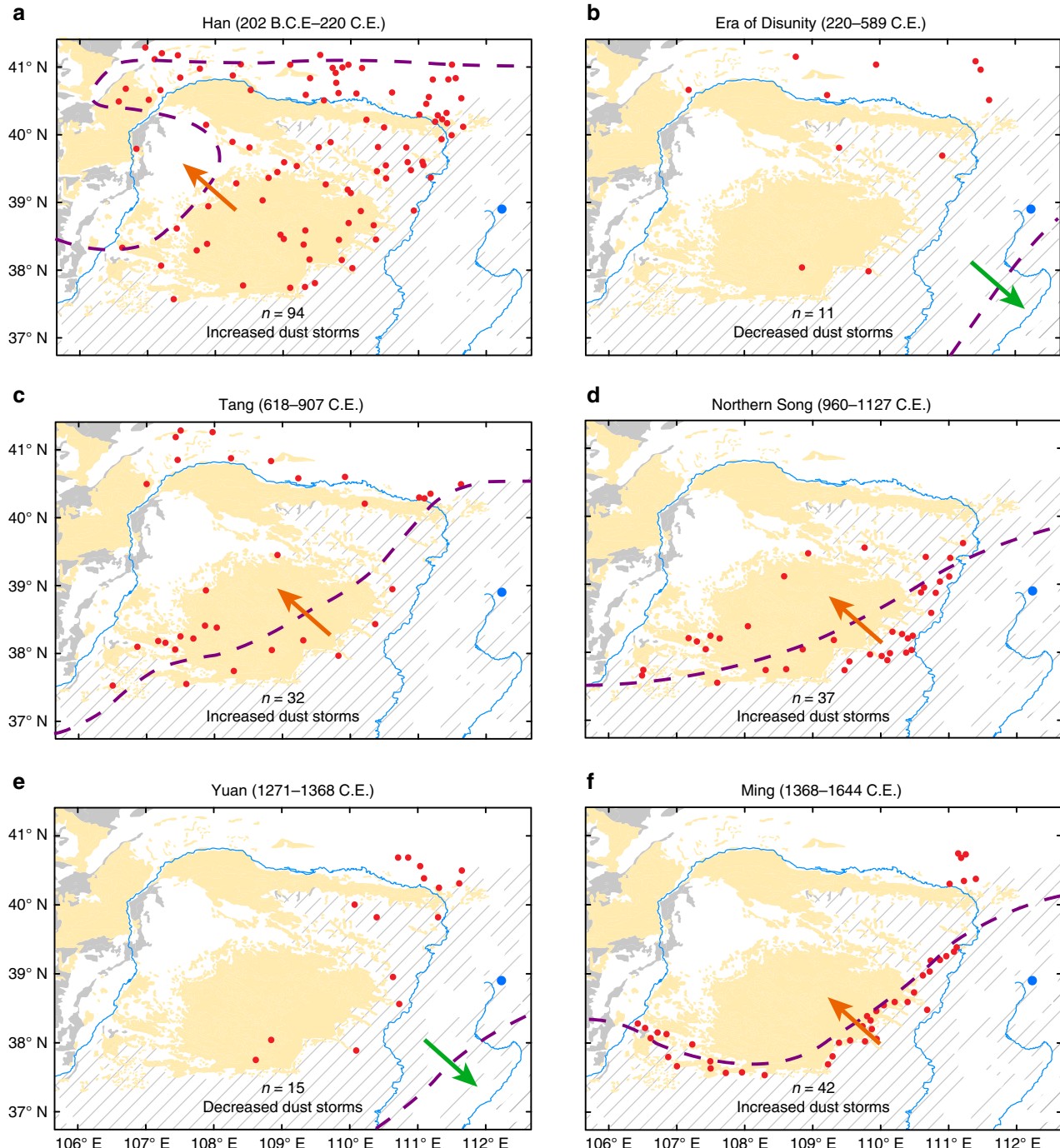

**Fig. 3 Variations of the farming-pastoral ecotone and spatial distribution of ancient cities in the dust source region from the Han to the Ming Dynasties. a, c, d, f** Periods of unified dynasties with an increased number of cities (red dots) in the dust source region, which correspond to the northward movement of the farming-pastoral ecotone (purple broken line). **b, e** Periods of civil unrest with fewer cities in the dust source region, which correspond to the southward movement of the farming-pastoral ecotone. Ancient cities are interpreted as the permanent settlements of agricultural populations[28]. This is because nomadic populations are highly mobile, preventing the construction of such settlements, whereas settled agricultural populations are forced by necessity to build cities. The arrows indicate the direction of movement of the establishment of ancient cities. Distributions of loess (gray line shading), desert (yellow shading), and Gobi (dark gray shading) are also indicated. The blue dot indicates the location of Lake Gonghai and the blue line indicates the Yellow River. The distribution of ancient cities and the farming-pastoral ecotone are modified from ref. [31] and ref. [32], respectively. The figure was created using Arcmap 10.2.

civil disunity and unrest, resulted in a retreat of the farming-pastoral ecotone[32] and the abandonment of ancient cities[31], such as occurred during the Era of Disunity and the Yuan Dynasty (Fig. 3b, e). Previously eroded soils were eventually stabilized by grassland regeneration[10,12]. This reduced the dust storm activity even against a climatic background of decreased monsoon rainfall (e.g., Five Dynasties and Ten Kingdoms period, 5D & 10K). These findings corroborate that anthropogenic factors surpassed climatic variability in controlling dust storms on multidecadal to centennial timescales in eastern China during the last 2250 years.

Notably, ancient Chinese society was complex and therefore other factors may potentially have contributed to the landscape changes. In the early stage of the Tang Dynasty, before 755 C.E., it appears that dust storm activity did not increase (Fig. 2c), likely because at that time the Tang Dynasty, unlike other unified dynasties, did not implement a policy of large-scale military farming in the dust source region[34]. After the coup of the "Rebellion of An Lu-Shan" (755–763 C.E.), in order to facilitate postwar reconstruction, the Tang Dynasty launched a tax-exemption policy to encourage farmers to cultivate underutilized land and pasture around the dust source region[12], causing a sharp increase in dust storm activity (Fig. 2c). This highlights the potential importance of government policy in shaping landscapes in China (see also Supplementary Note 1), although this remains to be further corroborated by finer temporal resolution records with a more reliable chronology.

AM variability played an important role in the rise and fall of unified dynasties in China[3,4]. Our results further demonstrate that the effect of intensified human activity during the unified dynasties outpaced natural climatic variability on earth surface processes as the major factor controlling dust storm activity, beginning at least 2000 years ago. Collectively, regardless of how the AM system changes in the future, intensified human activity without proper sustainable mitigation policies on land use will cause more severe dust storms by the accelerated expansion of drylands in China.

## Methods

**Sediment archives.** We collected two sediment cores (a long core and a short core) from alpine Lake Gonghai. The long core (GH09B) was obtained from the center of the lake (Supplementary Fig. 2a) in January of 2009 using a Uwitec Piston Corer. It was subsampled at 1 cm intervals, and here we focus on the uppermost lacustrine silty clay sediments (4.15 m), which span approximately the last 2000 years. The chronology was established using accelerator mass spectrometry $^{14}$C dating of eight terrestrial plant macrofossils samples, which are unaffected by the reservoir effect. Supplementary Table 1 lists the measured and calibrated ages. The ages are expressed in years before present (BP), where "present" is defined as 1950 C.E. Bayesian age-depth modeling was performed using OxCal v4.2.2 and a Poisson-process (P-sequence) single depositional model at 1 cm increments with a $K$ value of 100[35] to produce the age-depth model.

For comparison with instrumental records, the high-resolution short core (GH13F), 57.5 cm in length, was obtained from the deepest part of Lake Gonghai (Supplementary Fig. 2a) in August 2013 using a Universal gravity corer and sectioned on-site using a close-interval extruder into 0.5 cm intervals. The chronology of this high-resolution sediment core for the past ~175 years was established using a constant-rate-of-supply model applied to excess $^{210}$Pb inventories, counted on a digital, high-purity germanium spectrometer (DSPec, Ortec), following standard gamma counting procedures.

Ten surface lake sediment samples and thirteen ice-trapped dust samples were collected along the long axis of Lake Gonghai (Supplementary Fig. 2a). The surface lake sediments were obtained using the Universal gravity corer, and their uppermost 2 cm were used for grain-size analysis.

**Analytical methods.** For grain-size analysis, ~0.2 g samples were pretreated using the $H_2O_2$–HCl procedure, to remove organic matter and carbonates, respectively, and then dispersed with $(NaPO_3)_6$ and an ultrasonic shaker[36]. Grain-size frequency distributions were measured with a Malvern Mastersizer 2000 laser grain-size analyzer. Quartz grains were isolated by pretreating ~2 g samples with a $H_2O_2$–HCl–$K_2S_2O_7$–$H_2SiF_6$ digestion procedure[37]. The surface morphology of the quartz grains was observed using a Hitachi S4800 scanning electron microscope.

To determine the provenance of the CSC in the lake sediments, we measured the zircon U–Pb ages of bulk lake sediments, coarse silt particles, catchment surface bedrock, and surrounding loess. Detrital zircon grains were isolated using heavy liquid separation and were then selected randomly and analyzed by laser ablation inductively coupled plasma mass spectrometry. The laser-beam diameter was 30 μm. Glitter 4.4.2 was used to process the data[38], and the discordance values <10% (concordance >90% and <110%) were applied. We used $^{206}$Pb/$^{238}$U ages for zircon grains <1000 Ma, and $^{207}$Pb/$^{206}$Pb ages for zircon grains >1000 Ma. Around 120 zircon U–Pb ages for each sample were graphed in the form of a probability density plot and a Kernel Density Estimate[39].

**Mathematical partitioning of grain-size distributions.** The basic principle of this method is that a sedimentary environment, or any transport process, has a specific and stable grain-size distribution range. Thus, the grain-size range of the component can be used to indicate the frequency and intensity of the transport process. The procedure used was as follows.

First, the number of components was determined by the number of knee points of the frequency curve. The Weibull and Normal function were used to fit the measured grain-size data mathematically with a goodness-of-fit criteria. Second, the grain-size function formula was defined based on the component number and function type. After function fitting, the modal size and percentage of each component were calculated. Third, the origin of each component was identified based on the modal size, percentage, and comparison with the grain-size distribution of sediments of known origin, such as typical lake sediments[19,40] and Chinese loess[19,41,42].

**Interpretation of the origin of separated lake sediment grain-size components.** Using the grain-size distribution function method[19], five components (clay, fine silt, coarse silt, fine sand, and sand) were identified in the sediments from Lake Gonghai (Supplementary Fig. 4). Based on a comparison with grain-size distributions of sediments with known origin, and observation of the origins of the lake sediments, the transport processes associated with the various components were determined as follows.

The clay component originates from chemical weathering and pedogenesis of the catchment bedrock and is transported to the lake by surface runoff or by adhering to large grains during dust storms[43]. Fine silt is the dominant component of lake sediments (Supplementary Fig. 4d–f). Although it can be supplied by the dry and wet deposition of background dust[44], the major part is transported to the lake by surface runoff as an offshore-suspension component[40]. All of the sediments from Lake Gonghai possess this component, and it is often used to represent lacustrine sediments[19]. Coarse silt is the second dominant component (Fig. 1a and Supplementary Fig. 4), even though a few sediments do not consist of this component (Supplementary Fig. 4d). Thus, the number of this component is lower than that of fine silt component.

Only a few sediment samples contained fine sand or sand components (Supplementary Fig. 4f, g), and their percentages were <5%. Previous work demonstrated that sand or fine sand is a typical nearshore component and is deposited rapidly near the lake shore and cannot be transported to the lake center by surface runoff[40]. Furthermore, both sand and fine sand usually move by saltation across the land surface[45], and thus they cannot reach the lake center through direct air movement unless under highly optimal conditions[46]. Notably, the bedrock debris and ice-trapped dust are dominated by fine sand and sand components (Supplementary Fig. 2e, f). This suggests that these components in the lake sediments are mainly derived from the bedrock debris that was transported to the frozen lake surface during winter and subsequently supplied to the water column after ice melting.

## Data availability

The dust storm data were available at https://doi.org/10.6084/m9.figshare.11660679.

## Code availability

All C code used to separate grain-size components of sediments was based on the software "Sediment Component Analysis" developed by D.H.S. To request access to the code please contact D.H.S (dhsun@lzu.edu.cn).

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

## Acknowledgements

We thank D.H. Sun for grain-size partitioning of lake sediment; S.J. Gao for performing grain-size analyses of core GH09B; W. Huang, J.M. Sun, J.W. Zhang, J.S. Nie, W.F. Ruddiman, H.T. Wei, X.N. Zhang, H.C. Xie for valuable suggestions; S.P. Liu for analysing zircon U–Pb data; H. Azarmdel, Z.W. Shen for assistance in preparing the figures; J. Bloemendal for English improvement and discussion; and X.Y. Cao, D. Wu, Q. Wang for fieldwork, respectively. This work was supported by National Natural Science Foundation of China (Grants 41790421, 41722105).

## Author contributions

F.H.C. and J.B.L. designed the study. J.B.L., F.H.C., S.Q.C, X.Z., and J.P.S. led the interpretation and writing. Q.H.X., Z.L.W., J.H.C., and Y.C.L. led the field work. S.Q.C. carried out grain-size analysis and zircon U–Pb dating with guidance from X.W. and F.H.C. F.H.C., J.B.L., and J.H.C. generated the age model. S.Q.C, J.B.L., X.Z., and F.H.C. wrote the paper with substantial contributions from J.P.S., E.J.G., X.W., J.H.C., M.R.Q., G.H.D., and Y.Y.X. All the authors discussed and reviewed the paper prior to submission.

## Competing interests

The authors declare no competing interests.
