## [Peer Review File · Nature Communications]

Reviewers' comments:

Reviewer #1 (Remarks to the Author):

The manuscript entitled "Asian dust-storm activity dominated by Chinese dynasty changes since 2000 BP" presents empirical evidence to illustrate which factor mainly contributes to the frequency of dust storms in Chinese history. Besides, it is further concluded that human activities overtake natural climatic variability to determine dust storm incidents in eastern China. The merits of this study are: First, a 2250-year long dust storm intensity in China is reconstructed, which is a valuable knowledge contribution to academia. Second, the significance of human activities in distributing the physical environment in the historical period has been demonstrated, which provides important insight into the investigation of environmental sustainability in the past. Briefly, this is a very good interdisciplinary study illustrating the connection between the physical environment and human societies. Both the academic people and the lay public will be interested in this piece of work. Given the important contribution of this study, I recommend it to be accepted for publication.

I have a few suggestions/comments to the manuscript, which are listed below:

Could further analysis (such as time-series analysis) be conducted to measure the respective effect of population growth and climate change on the frequency of dust storm activities?

Human activities are also determined by natural climatic variability. Besides, population growth, climate change, and land degradation are inter-correlated. For instance, climate change and population growth can cause land degradation. When the land is degraded, people will leave, resulting in the feedback loop between land degradation and population growth, just like a chicken and egg situation. This is especially true at the multi-decadal to centennial time scale. In this study, when it is stated that human activities overtake natural climatic variability to determine dust storm incidents in eastern China, is it something at the multi-centennial to millennial time scale? It will be good if this point could be clarified in the main text.

In lines 159–163, it is stated that as the pronounced increase in dust storm activity is unrelated to wind strength and hence, climatic factors played a limited role in modulating dust storm activity. But, wind strength is only one of the climatic indicators, while the happening of the dust storm is also contingent upon the aridity thresholds. Therefore, the above argument may need to be revised.

Regarding Figure 3, are those ancient "cities" simply human settlements? Please clarify. Besides, could those "cities" be further categorized as agricultural, agri-nomad, or nomad according to their subsistence strategies? Such categorization may be helpful when discussing the influence of human activities on the frequency of dust storms.

Reviewer #2 (Remarks to the Author):

Review of manuscript NCOMMS-19-25612-T, entitled: "Asian dust-storm activity dominated by Chinese dynasty changes since 2000 BP", which was submitted for publication in Nature Communications by Fahu Chen, Shengqian Chen, Xu Zhang, Jianhui Chen, Xin Wang, Evan J. Gowan, Mingrui Qiang, Guanghui Dong, Zongli Wang, Yuecong Li, Qinghai Xu, Yangyang Xu, John P. Smol, William F. Ruddiman and Jianbao Liu.

The manuscript describes a study of desert dust deposited in a small Alpine lake at the edge of the Chinese Loess Plateau. A highly-resolved sediment core allows a reconstruction of dust input into the lake since the last ~2kyr and which shows increases in dust deposition that line up with dramatic changes in Chinese population dynamics.

The manuscript is easy-to-read, constructed well and apart from some minor comments I would recommend it to be accepted for publication in Nature Communications as it nicely shows how population dynamics, land-use change and desertification interact. The authors themselves conclude that "without proper sustainable mitigation policies on land use will cause more severe dust storms by the accelerated expansion of drylands in China" With their records, the authors show how human influence on dust-storm activity is presently larger than the immense Asian-Monsoon system.

Minor comments

L127-129, I am not convinced by the "eolian surface features", please elaborate?

L157-159, if I understood correctly, this is also what Stefan Mulitza and co-workers showed in their 2010 paper (the authors' reference #17) have demonstrated as well.

L274, please provide the number of particle-size analyses?

Supplementary Info; the authors cite Kenneth Pye's 1987 paper in which he presents a model for dust transport in which he argues that large particles (sand, >63µm) cannot possibly be transported long distances through the air. However, recently, Michèlle van der Does and co-workers (<https://doi.org/10.1126/sciadv.aau2768>) have demonstrated that this is not the case and so-called "giant particles" are able to travel long distances (> 1000 km) through the air. Given the size of Lake Gonghai, it should be no problem for sand particles to be deposited in the center of the lake. Nonetheless, I agree with the authors that the coarse-silt fraction is most likely carried to the lake by aeolian transport and that it can be interpreted in terms of dust mobilisation, which is due to removal of vegetation by human land use.

Figure S6 (mentioned in L127-129) I am not convinced by the eolian features; please elaborate? I appreciate the authors' effort to find additional evidence for their proxy but I would have believed their story without these SEM photographs.

The same holds for Figure S7; the comparison of their dust-storm proxy with non-sea salt ions in the GISP2 ice core. Some parts of the records indeed line up but if these records can be compared one-to-one, why doesn't the GISP2 record reflect dustiness in the period 750-1250 C.E.? As in Figure S6, I would have believed the authors' story without this comparison.

Typos

L62, remove "The" before China.

L87, "systemic" should be "systematic"

L90, don't capitalise Earth

L109, replace "enables" with "allows"

L198, don't capitalise Earth

L343, "modal" should be "model"

In conclusion, I quite like this manuscript and would recommend it to be accepted for publication in Nature Communications.

Best wishes,

Jan-Berend Stuut

Reviewer #3 (Remarks to the Author):

I find the data collection, analysis, and interpretation of the results to be suitable and effective. They are well presented and easily followed. I feel that the authors have done a good job presenting and interpreting the data.

My concerns are relatively minimal and have only to do with how the data are interpreted in a historical context.

One thing that might be re-thought is the thrust of the abstract and the paper. The authors begin by asking if human activity could overtake Asian Monsoon (AM) variability. In the end, though, the focus is on dust storm activity, not specifically the anthropogenic (Anthropocene) effect on AM. There is a subtle difference that is being elided with the author's approach. There still isn't evidence that this "early" Anthropocene affected AM activity. The feedback loop between AM and dust storms is still unknown. I think the author's have made an excellent case for an anthropogenic footprint in dust storm activity. My point is simply that they've not yet made a link to how this dust storm activity influences the AM. This sort of causal link emerges as one of the great issues of the entire "Anthropocene debate," which is why I believe it is important to be very clear about how we are presenting the argument.

The authors use population data to make their case for the relationship between dust storms and dynastic histories. These population data are important, but I think the authors appreciate that these population histories, like any data source, come with several interpretive challenges. These data problems are especially notable in pre-Han times, as well as in times of political unrest, where the capacity to collect census data is minimized. My suggestion is that the author's either move away from making absolute statements about population, or they at least qualify the data by indicating that there is uncertainty in these counts. This paper will make an important contribution, but uncritical acceptance of data does not help advance the argument.

Similarly, I'm not especially confident in the cause-and-effect linkage between AM fluctuations and the integrity of various dynasties. A cursory reading of Chinese history suggests that there are very complicated reasons for the evolution of various dynasties. The author's make this case, in fact, when they mention the An-Shi (An Lushan) rebellion and its aftermath. The revised tax policies in Tang times generated a new pattern that involved repopulation of parts of the loess plateau. The AM had no clearly connected relationship to the pattern noted here in this paper. The point being that the patterns we see are remarkably complex and depend on a host of variables, of which climate, AM modulation, dust storms, and politics are only some of the factors. I encourage the authors to please take note of this complexity so that these data are not misinterpreted as being absolute—they really are not.

Finally, I'm still not clear if humans caused an increase in dust storm "activity" or if the human footprint (agriculture, deforestation- which isn't mentioned as an issue but might be??— technological change, and desertification), affected the capacity of dust storms to mobilize and transport sediments. I'd encourage the author's to more directly address this issue.

Tristram R. Kidder

Responses to reviewer's comments:

The responses are in blue. The revisions are marked in red in the revised manuscript.

The comments were separated into several parts and responded to point by point.

Reviewer #1 (Remarks to the Author):

The manuscript entitled “Asian dust-storm activity dominated by Chinese dynasty changes since 2000 BP” presents empirical evidence to illustrate which factor mainly contributes to the frequency of dust storms in Chinese history. Besides, it is further concluded that human activities overtake natural climatic variability to determine dust storm incidents in eastern China. The merits of this study are: First, a 2250-year long dust storm intensity in China is reconstructed, which is a valuable knowledge contribution to academia. Second, the significance of human activities in distributing the physical environment in the historical period has been demonstrated, which provides important insight into the investigation of environmental sustainability in the past. Briefly, this is a very good interdisciplinary study illustrating the connection between the physical environment and human societies. Both the academic people and the lay public will be interested in this piece of work.

Given the important contribution of this study, I recommend it to be accepted for publication.

I have a few suggestions/comments to the manuscript, which are listed below:

Could further analysis (such as time-series analysis) be conducted to measure the respective effect of population growth and climate change on the frequency of dust storm activities?

Response: The suggestion was carefully considered. We agree with you that it would be better if we could distinguish the relative contributions of population growth and climate change. However, this paper focuses on the question of whether intensified human activity during the unified dynasties outpaced natural climatic variability on Earth surface processes as the major factor controlling dust storm activity. Determination of the respective effects of population growth and climate change is beyond the scope of this paper. However, we will conduct further

analysis in the future because addressing this issue requires a specific model.

Human activities are also determined by natural climatic variability. Besides, population growth, climate change, and land degradation are inter-correlated. For instance, climate change and population growth can cause land degradation. When the land is degraded, people will leave, resulting in the feedback loop between land degradation and population growth, just like a chicken and egg situation. This is especially true at the multi-decadal to centennial time scale. In this study, when it is stated that human activities overtake natural climatic variability to determine dust storm incidents in eastern China, is it something at the multi-centennial to millennial time scale? It will be good if this point could be clarified in the main text.

Response: Thank you for your affirmation of our work. Our records indeed indicate that human activity overtook natural climatic variability as the dominant control of dust storms on multi-decadal to centennial timescales. However, the occurrence of dust storms on multi-centennial to millennial is insufficiently discussed in this paper due to the limited time span of the record. Therefore, according to your suggestion, we have highlighted this in the main text - please see Lines 169, 196.

In lines 159–163, it is stated that as the pronounced increase in dust storm activity is unrelated to wind strength and hence, climatic factors played a limited role in modulating dust storm activity. But, wind strength is only one of the climatic indicators, while the happening of the dust storm is also contingent upon the aridity thresholds. Therefore, the above argument may need to be revised.

Response: Thank you for your comments. We have stated the positive correlation between monsoon rainfall and dust storm in the manuscript, indicating that the direct effects of monsoon rainfall on dust storm were weak (please see Lines 165-167). Together with the weak correlation between wind speed and dust storms, we therefore concluded that climatic factors played a limited role in modulating dust storm activity. In order to make this clearer, we have revised the sentence - please see Line 166.

Regarding Figure 3, are those ancient “cities” simply human settlements? Please clarify. Besides, could those “cities” be further categorized as agricultural, agri-nomad, or nomad according to their subsistence strategies? Such categorization may be helpful when discussing the influence of human activities on the frequency of dust storms.

Response: Yes, the ancient cities in Fig. 3 are all human settlements. We have clarified this in Lines 472-475. Furthermore, all of these cities are categorized as agricultural (Feng, 2008). The reason for this is that nomadic populations are highly mobile which prevents their building cities. Settled agricultural populations, in contrast, need to build cities. Therefore, we did not further categorize these cities.

Feng, W. Y., 2008. The study on the historical cities of the Ordos Plateau and its periphery in North China. (Doctoral dissertation) Lanzhou University, Lanzhou.

Reviewer #2 (Remarks to the Author):

Review of manuscript NCOMMS-19-25612-T, entitled: “Asian dust-storm activity dominated by Chinese dynasty changes since 2000 BP”, which was submitted for publication in Nature Communications by Fahu Chen, Shengqian Chen, Xu Zhang, Jianhui Chen, Xin Wang, Evan J. Gowan, Mingrui Qiang, Guanghui Dong, Zongli Wang, Yuecong Li, Qinghai Xu, Yangyang Xu, John P. Smol, William F. Ruddiman and Jianbao Liu.

The manuscript describes a study of desert dust deposited in a small Alpine lake at the edge of the Chinese Loess Plateau. A highly-resolved sediment core allows a reconstruction of dust input into the lake since the last ~2kyr and which shows increases in dust deposition that line up with dramatic changes in Chinese population dynamics.

The manuscript is easy-to-read, constructed well and apart from some minor comments I would recommend it to be accepted for publication in Nature Communications as it nicely shows how

population dynamics, land-use change and desertification interact. The authors themselves conclude that “without proper sustainable mitigation policies on land use will cause more severe dust storms by the accelerated expansion of drylands in China” With their records, the authors show how human influence on dust-storm activity is presently larger than the immense Asian-Monsoon system.

Minor comments

L127-129, I am not convinced by the “eolian surface features”, please elaborate?

Response: We appreciate your constructive suggestion. We now provide new, more detailed Scanning Electron Microscope (SEM) photographs of isolated quartz particles in Supplementary Fig. 6. The characteristic textures include dish-shaped concavities, mechanically formed upturned plates, elongated depressions, smooth precipitation surfaces, cleavage faces and arcuate fractures (Supplementary Fig. 6b–f). These textural characteristics indicate an eolian origin (Krinsley and Trusty, 1985; Licht et al., 2014; Vos et al., 2014). Although the quartz particles potentially may be affected by other processes, the original grain-size properties are retained. Thus we suggest that the main body of the coarse silt particles are of eolian origin. Please see Lines 128-132 and revised Supplementary Fig. 6.

Krinsley, D., Trusty, P., 1985 Environmental interpretation of quartz grain surface textures. In: Zuffa, G. G. (Ed.), Provenance of Arenites, Springer Netherlands, pp. 213–229.

Vos, K., et al., 2014. Surface textural analysis of quartz grains by scanning electron microscopy (SEM): From sample preparation to environmental interpretation. Earth-Sci. Rev. 128, 93–104.

Licht, A. et al., 2014. Asian monsoons in a late Eocene greenhouse world. Nature 513, 501–506.

L157-159, if I understood correctly, this is also what Stefan Mulitza and co-workers showed in their 2010 paper (the authors’ reference #17) have demonstrated as well.

Response: Agreed. We have added this reference.

L274, please provide the number of particle-size analyses?

Response: Agreed. Revised as suggested.

Supplementary Info; the authors cite Kenneth Pye's 1987 paper in which he presents a model for dust transport in which he argues that large particles (sand, $>63\mu\text{m}$) cannot possibly be transported long distances through the air. However, recently, Michèlle van der Does and co-workers (<https://doi.org/10.1126/sciadv.aau2768>) have demonstrated that this is not the case and so-called "giant particles" are able to travel long distances (> 1000 km) through the air. Given the size of Lake Gonghai, it should be no problem for sand particles to be deposited in the center of the lake. Nonetheless, I agree with the authors that the coarse-silt fraction is most likely carried to the lake by aeolian transport and that it can be interpreted in terms of dust mobilisation, which is due to removal of vegetation by human land use.

Response: We appreciate your attention to detail and your constructive suggestion. We have added this point, please see Lines 286-288.

Figure S6 (mentioned in L127-129) I am not convinced by the eolian features; please elaborate? I appreciate the authors' effort to find additional evidence for their proxy but I would have believed their story without these SEM photographs.

Response: As stated above, we have elaborated the surface features of isolated quartz particles in the manuscript. Please see Lines 128-132 and revised Supplementary Fig. 6.

The same holds for Figure S7; the comparison of their dust-storm proxy with non-sea salt ions in the GISP2 ice core. Some parts of the records indeed line up but if these records can be compared one-to-one, why doesn't the GISP2 record reflect dustiness in the period 750-1250 C.E.? As in Figure S6, I would have believed the authors' story without this comparison.

Response: The non-sea salt ions from GISP2 is an indicator of the intensity of Siberian High (Mayewski et al., 2004; Mayewski and Maasch, 2006) that is widely accepted to dominate the wind speed in dust source regions during winter and spring. The comparison in Fig. S7 is meant to investigate the possible influence of wind speed on dust storm activity. Just as you said that

the dust storm record is not in accord with the GISP2 record, we in fact use this result to conclude that wind speed is not the dominant factor controlling dust storms. Therefore, the fact that the GISP2 record does not reflect dustiness during the period of 750-1250 C.E. supports our conclusion.

Mayewski, P. A. et al., 2014. Holocene climate variability. Quat. Res. 62, 243–255.

Mayewski, P. A., Maasch, K. A., 2006. Recent warming inconsistent with natural association between temperature and atmospheric circulation over the last 2000 years. Clim. Past Discuss. 2, 327–355.

Typos

L62, remove “The” before China.

Response: Agreed, revised as suggested.

L87, “systemic” should be “systematic”

Response: Agreed, revised as suggested.

L90, don’t capitalise Earth

Response: Agreed, revised as suggested.

L109, replace “enables” with “allows”

Response: Agreed, revised as suggested.

L198, don’t capitalise Earth

Response: Agreed, revised as suggested.

L343, “modal” should be “model”

Response: Agreed, revised as suggested.

In conclusion, I quite like this manuscript and would recommend it to be accepted for

publication in Nature Communications.

Best wishes,

Jan-Berend Stuut

Reviewer #3 (Remarks to the Author):

I find the data collection, analysis, and interpretation of the results to be suitable and effective. They are well presented and easily followed. I feel that the authors have done a good job presenting and interpreting the data.

My concerns are relatively minimal and have only to do with how the data are interpreted in a historical context.

One thing that might be re-thought is the thrust of the abstract and the paper. The authors begin by asking if human activity could overtake Asian Monsoon (AM) variability. In the end, though, the focus is on dust storm activity, not specifically the anthropogenic (Anthropocene) effect on AM. There is a subtle difference that is being elided with the author's approach. There still isn't evidence that this "early" Anthropocene affected AM activity. The feedback loop between AM and dust storms is still unknown. I think the authors have made an excellent case for an anthropogenic footprint in dust storm activity. My point is simply that they've not yet made a link to how this dust storm activity influences the AM. This sort of causal link emerges as one of the great issues of the entire "Anthropocene debate," which is why I believe it is important to be very clear about how we are presenting the argument.

Response: Thanks for your comments. Dust storms are caused by wind erosion of surface soil, which is controlled by human activity and/or climatic conditions. Based on our records, we found the dust storm variations during the past ~2000 years were dominated by human activity (e.g. agriculture, deforestation), while the climate conditions (i.e. the AM and wind speed)

played a limited role. We thus concluded that the impact of human activity has overtaken that of the AM on landscape changes. This is the focus of this paper, rather than the feedback on the AM.

With regard to the feedback of dust storms on the AM, we suggest that the AM system is a climate state that has persisted for a long time, which thus is able to affect the vegetation cover and finally influence the occurrence of dust storms. In contrast, a dust storm is a short time atmospheric event that is discontinuous, and therefore is difficult for them to modulate a long-standing atmospheric circulation such as the AM system. However, we agree that dust storms can affect local/regional precipitation or a single rainfall process, and we intend to carry out this work in the future using modelling.

The authors use population data to make their case for the relationship between dust storms and dynastic histories. These population data are important, but I think the authors appreciate that these population histories, like any data source, come with several interpretive challenges. These data problems are especially notable in pre-Han times, as well as in times of political unrest, where the capacity to collect census data is minimized. My suggestion is that the author's either move away from making absolute statements about population, or they at least qualify the data by indicating that there is uncertainty in these counts. This paper will make an important contribution, but uncritical acceptance of data does not help advance the argument.

Response: Thank you for pointing this out. According to your suggestion, we have deleted the absolute statements about the population. We agree with you that the reliability of the population record is of great important in this work. In this paper, we focus on relative changes in population (growth/decrease). An absolute statement about population will be ambiguous, however, the relative changes in population are highly reliable. For instance, the population would increase after the establishment of a unified dynasty, while the population would decrease during periods of civil unrest (Zhao and Xie, 1988). The main conclusion of this paper is based on relative changes in population.

Zhao, W. L., Xie, S. J., 1988. History of population in China. People's Publishing House, Beijing.

Similarly, I'm not especially confident in the cause-and-effect linkage between AM fluctuations and the integrity of various dynasties. A cursory reading of Chinese history suggests that there are very complicated reasons for the evolution of various dynasties. The author's make this case, in fact, when they mention the An-Shi (An Lushan) rebellion and its aftermath. The revised tax policies in Tang times generated a new pattern that involved repopulation of parts of the loess plateau. The AM had no clearly connected relationship to the pattern noted here in this paper. The point being that the patterns we see are remarkably complex and depend on a host of variables, of which climate, AM modulation, dust storms, and politics are only some of the factors. I encourage the authors to please take note of this complexity so that these data are not misinterpreted as being absolute—they really are not.

Response: We fully agree with you that it would be too simplistic to speculate that all dynastic changes are driven by climatic events. We have added a qualifying statement which emphasizes that we do not consider these data to be absolute - please see Lines 198-199. From current evidence we only can propose that climate, human activities, or politics are more important factors affecting dust storm, although we cannot exclude other variables. However, we believe this conclusion is generally reliable if all of the different variables match well on a long timescale, because there are always existed some relationships between them.

Finally, I'm still not clear if humans caused an increase in dust storm "activity" or if the human footprint (agriculture, deforestation- which isn't mentioned as an issue but might be??— technological change, and desertification), affected the capacity of dust storms to mobilize and transport sediments. I'd encourage the author's to more directly address this issue.

Response: Good point. In this paper, we consider human activity as the human footprint (e.g., agriculture, deforestation, technological change, and desertification). We have clarified this - please see lines 171-172.

Tristram R. Kidder

REVIEWERS' COMMENTS:

Reviewer #1 (Remarks to the Author):

All of my previous comments have been addressed by the authors. Therefore, I recommend this manuscript be accepted for publication.